# Flavivirus NS1 Triggers Tissue-Specific Disassembly of Intercellular Junctions Leading to Barrier Dysfunction and Vascular Leak in a GSK-3β-Dependent Manner

**DOI:** 10.3390/pathogens11060615

**Published:** 2022-05-24

**Authors:** Henry Puerta-Guardo, Scott B. Biering, Francielle Tramontini Gomes de Sousa, Jeffrey Shu, Dustin R. Glasner, Jeffrey Li, Sophie F. Blanc, P. Robert Beatty, Eva Harris

**Affiliations:** 1Division of Infectious Diseases and Vaccinology, School of Public Health, University of California, Berkeley, CA 94720-3370, USA; sbiering@berkeley.edu (S.B.B.); francielletg@gmail.com (F.T.G.d.S.); jeffrey.l.shu@gmail.com (J.S.); drglasner@gmail.com (D.R.G.); jeff.li@berkeley.edu (J.L.); sophieblanc@berkeley.edu (S.F.B.); prbeatty@berkeley.edu (P.R.B.); 2Laboratorio de Virologia, CIR-Biomedicas y Unidad Colaborativa de Bioensayos Entomologicos (UCBE), Universidad Autonoma de Yucatan, Merida 97000, Mexico

**Keywords:** flavivirus, NS1 protein, intercellular junction complex, adherens junction, β-catenin, VE-cadherin, phosphorylation, GSK-3β, endothelial hyperpermeability, vascular leakage

## Abstract

The flavivirus nonstructural protein 1 (NS1) is secreted from infected cells and contributes to endothelial barrier dysfunction and vascular leak in a tissue-dependent manner. This phenomenon occurs in part via disruption of the endothelial glycocalyx layer (EGL) lining the endothelium. Additionally, we and others have shown that soluble DENV NS1 induces disassembly of intercellular junctions (IJCs), a group of cellular proteins critical for maintaining endothelial homeostasis and regulating vascular permeability; however, the specific mechanisms by which NS1 mediates IJC disruption remain unclear. Here, we investigated the relative contribution of five flavivirus NS1 proteins, from dengue (DENV), Zika (ZIKV), West Nile (WNV), Japanese encephalitis (JEV), and yellow fever (YFV) viruses, to the expression and localization of the intercellular junction proteins β-catenin and VE-cadherin in endothelial cells from human umbilical vein and brain tissues. We found that flavivirus NS1 induced the mislocalization of β-catenin and VE-cadherin in a tissue-dependent manner, reflecting flavivirus disease tropism. Mechanistically, we observed that NS1 treatment of cells triggered internalization of VE-cadherin, likely via clathrin-mediated endocytosis, and phosphorylation of β-catenin, part of a canonical IJC remodeling pathway during breakdown of endothelial barriers that activates glycogen synthase kinase-3β (GSK-3β). Supporting this model, we found that a chemical inhibitor of GSK-3β reduced both NS1-induced permeability of human umbilical vein and brain microvascular endothelial cell monolayers in vitro and vascular leakage in a mouse dorsal intradermal model. These findings provide insight into the molecular mechanisms regulating NS1-mediated endothelial dysfunction and identify GSK-3β as a potential therapeutic target for treatment of vascular leakage during severe dengue disease.

## 1. Introduction

The *Flavivirus* genus is composed of enveloped, positive-sense RNA viruses, including medically important mosquito-transmitted viruses such as dengue (DENV), Zika (ZIKV), West Nile (WNV), Japanese encephalitis (JEV), and yellow fever (YFV) viruses, and viruses transmitted by ticks such as tick-borne encephalitis virus (TBEV) [1]. Human infections with flaviviruses lead to a wide range of outcomes and disease manifestations, from asymptomatic infections to severe life-threatening disease characterized by vascular leakage, hemorrhage, organ failure, neurological disorders, and/or fetal abnormalities [2,3].

The flavivirus genome encodes three structural (C, prM/M, and E) and seven nonstructural proteins (NS1, NS2A, NS2B, NS3, NS4A, NS4B, and NS5) [4]. NS1 is essential for flavivirus replication and is found intracellularly as a dimer associated with the replication complex on the cytoplasmic side of the endoplasmic reticulum and on membranes at the cell surface. NS1 also forms a hexameric barrel-shaped protein that is secreted into the extracellular space from infected cells [5]. Secreted NS1 has been implicated in a variety of processes, including immune evasion and pathogenesis via modulation of the complement cascade [6].

We and others have shown that soluble flavivirus NS1 also directly triggers pathogenesis both in vitro and in vivo via degradation of the endothelial glycocalyx layer (EGL) [7,8,9,10,11]. The EGL is a complex membrane-bound matrix of carbohydrates, proteoglycans, and glycoproteins that lines the luminal surface of the vascular endothelium, with multiple roles in vascular homeostasis [12]. It regulates endothelial permeability and the interactions between the endothelium and blood cells, and it endures the mechanical forces of the bloodstream, protecting the endothelium against laminar shear stress [12,13,14]. Moreover, EGL disruption has been observed in humans with acute dengue, and levels of shed EGL components, such as heparan sulfate (HS) [15,16], chondroitin sulfate (CS) [15], hyaluronic acid [17,18], and syndecan-1 and endocan [17], have been associated with plasma leak severity and NS1 levels [18]. NS1 triggers EGL degradation by activating multiple enzymes in endothelial cells (EC), including neuraminidases, cathepsin L, and heparanase. These enzymes cleave EGL components on the cell surface in a cytokine-independent manner, which contributes to endothelial hyperpermeability and vascular leak [7,8,10,11,19,20,21,22]. We have previously shown that this endothelial barrier impairment induced by different flavivirus NS1 proteins occurs in a tissue-specific manner, reflecting the tropism and disease manifestations of the corresponding flavivirus. For example, NS1 from ZIKV triggers barrier dysfunction in umbilical vein ECs, placental explants, and brain ECs, while NS1 from DENV, a pantropic flavivirus, causes EC dysfunction in lung, brain, umbilical vein, liver, and skin; YFV NS1 exerts the most significant effect on liver ECs; and the neurotropic flaviviruses, WNV and JEV, cause hyperpermeability and EGL disruption in ECs originating from human brain [19,22,23]. This same pattern of organ-specific leak was observed upon inoculation of mice with NS1 proteins from the different flaviviruses [19].

In addition to the EGL, intercellular junctions (IJCs), including tight and adherens junctions (TJ/AJ), are a major determinant of endothelial barrier function [24,25,26]. TJ/AJ proteins help control paracellular permeability by providing a semipermeable barrier to fluids, ions, and solutes. TJ/AJ proteins also function as signaling hubs, where transmembrane proteins of the TJ (occludin and claudin) and AJ (cadherin) are anchored by cytoplasmic scaffolding proteins such as zona occludens (ZO) and catenins, respectively, which then transmit signals intracellularly [25]. The AJ proteins, composing the cadherin/catenin adhesion system, are key regulators of tissue architecture and dynamics via control of cell proliferation, polarity, shape, motility, and survival [25,26]. Thus, their dysregulation and disruption in different tissues are linked to pathological abnormalities and disease [27,28,29,30,31]. β-catenin is a multifunctional protein of the cadherin/catenin adhesive complex through which cells adhere to one another. It binds to the cytoplasmic tail of cadherins—single-pass transmembrane glycoproteins found in many tissues, including the vascular endothelium (VE-cadherin) [32]. Thus, any protein modification that regulates the β-catenin/cadherin interaction significantly impacts cell–cell adhesion [29]. Although the interaction between cadherins and β-catenin is relatively stable, β-catenin levels are tightly regulated via a cascade of phosphorylation events mediated by cellular kinases (e.g., CK1, GSK-3) that result in modulation of cadherin/β-catenin binding interactions [33]. One of the major kinases that phosphorylates β-catenin is GSK-3β [34]. GSK3 is a serine/threonine protein kinase with two isoforms (GSK3α and GSK3β) that modulates the localization of β-catenin to either the AJ in its non-phosphorylated state or the cytoplasm/nucleus in its phosphorylated state [35,36,37,38]. Phosphorylation of β-catenin starts at serine residue 45 (S45) by casein kinase Iα, followed by phosphorylation by GSK3β at two other serine residues, S33 and S37, as well as one threonine residue, T41 [33,34].

DENV NS1 has been shown to modulate the permeability as well as the TJ/AJ proteins (e.g., ZO-1, VE-cadherin) of human ECs from skin (HMEC-1) [21] and umbilical vein (HUVEC) [39,40]; however, the effect of NS1 from other flaviviruses on the integrity of IJCs and the cellular mechanisms leading to their disruption remain obscure. Given the central role of IJCs in preserving the homeostasis of the EC barrier, here we examined the effect of NS1 from five distinct flaviviruses (DENV, ZIKV, WNV, JEV, and YFV) on the fate of VE-cadherin and β-catenin within endothelial cell-to-cell contacts. To do so, we used cultures of human ECs in vitro along with a trans-endothelial electrical resistance (TEER) assay as a proxy for the endothelium that lines the inner face of blood vessels in different tissues and its barrier function, together with a fluorescence-based approach to quantify vascular leak in vivo. Furthermore, we identified a requirement for GSK-3β in NS1-mediated endothelial hyperpermeability and vascular leak, as demonstrated using a GSK-3β-specific inhibitor. Taken together, this study provides new molecular insights into the tissue-specific mechanisms by which flavivirus NS1 proteins trigger the disruption of IJCs and identifies GSK-3β as a host kinase that may be targeted for treatment of pan-flavivirus NS1-mediated pathology.

## 2. Results

### 2.1. Flavivirus NS1 Proteins Alter the Localization of the AJ Proteins VE-Cadherin and β-Catenin in Human Endothelial Cells from the Brain and Umbilical Vein

In previous work, we demonstrated that secreted NS1 proteins from distinct flaviviruses can modulate the permeability of human ECs and cause increased vascular leak in a tissue-specific manner, correlating with disease tropism of the respective virus [19,22]. We have also shown that this phenomenon depends on the activation of cellular enzymes (sialidases, heparanase, and cathepsin L), resulting in the disruption of the EGL and contributing to endothelial hyperpermeability [8]. Here, we tested whether NS1 activates pathways regulating the architecture of the IJC. To do this, we examined the effect of different flavivirus NS1 proteins on the cellular distribution of VE-cadherin and β-catenin in human ECs. We found that NS1 treatment of human umbilical vein (HUVEC) and brain microvascular (HBMEC) ECs resulted in noticeable changes in the localization/distribution of VE-cadherin and β-catenin compared to the untreated controls, in which the distribution of these proteins adopt a “chicken fence” staining pattern outlining the cell–cell junctions (Figure 1A,B). As expected, the staining for VE-cadherin and β-catenin resulted in a continuous pattern outlining the cell borders in the untreated ECs (HUVEC and HBMEC controls). Six hours post-treatment (hpt) with DENV or ZIKV NS1 proteins (5 μg/mL), altered staining of both proteins in the HUVEC monolayers was observed, as indicated by the arrows, whereas treatment with WNV, JEV, and YFV NS1 did not induce noticeable changes (Figure 1A,B, top panel). In contrast, the staining pattern for both proteins was altered in HBMEC monolayers exposed to NS1 proteins from DENV, ZIKV, WNV, and JEV, but not YFV (Figure 1A,B, lower panel). In both human ECs, NS1-specific treatment led to changes in the distribution of VE-cadherin and β-catenin, suggesting their mislocalization from the IJC. These tissue-specific patterns of IJC disruption are consistent with the tissue-specific tropism of flavivirus NS1 proteins observed previously [19]. Although we observed a reorganization of the cellular localization of these IJC proteins, the mean fluorescent intensity (MFI) values, used as a proxy of total IJC protein expression, were not significantly altered (Figure 1C–F). In agreement, we detected no significant modulation of the VE-cadherin and β-catenin protein expression levels in ECs treated with flavivirus NS1 proteins by Western blot analyses of the cell lysates (Figure 2A–D). Taken together, these data suggest that in vitro NS1 treatment results in mislocalization but not degradation of IJC proteins VE-cadherin and β-catenin in HUVEC and HBMEC monolayers.

### 2.2. NS1 Triggers Endocytosis of VE-Cadherin and Phosphorylation of β-Catenin in Human Endothelial Cells

Next, we investigated the mechanism(s) by which flavivirus NS1 proteins trigger IJC mislocalization in EC monolayers. We first monitored the internalization of VE-cadherin, focusing on DENV and ZIKV NS1 treatment of HUVEC monolayers (Figure 3A,B). We observed increased colocalization of VE-cadherin with clathrin heavy chain proteins, indicating the potential formation of endocytic clathrin-coated pits, in HUVECs 6 hpt with DENV or ZIKV NS1 proteins, compared to untreated cells or another related flavivirus NS1 protein from WNV (Figure 3A). Quantification of co-staining of VE-cadherin and clathrin showed increased numbers of colocalized puncta in HUVEC monolayers treated with DENV and ZIKV NS1 compared to untreated cells or WNV NS1-treated cells (Figure 3B). Additionally, we examined the phosphorylation status of β-catenin 6 hpt with NS1 proteins from DENV, ZIKV, WNV, JEV, and YFV using a commercial cell-based ELISA to determine the level of expression and phosphorylation, as well as the percentage of β-catenin phosphorylated at serine residue 45 (S45) (Figure 3C,D). We also utilized a pan-phosphorylation polyclonal antibody that can detect β-catenin phosphorylation at serine 33, serine 37, or threonine 41 (S33/S37/T41) by IFA (Figure 3E). ELISA results showed that DENV and ZIKV NS1, but not WNV, JEV, or YFV NS1, induced a significant increase in phosphorylation of β-catenin at Ser-45 in HUVECs, in comparison to the untreated controls (Figure 3C,D). Additionally, IFA analysis revealed that treatment of HUVECs with DENV and ZIKV NS1, but not with WNV, JEV, or YFV NS1, led to increased phosphorylation of endogenous levels of β-catenin at any of the three residues S33/S37/T41 (Figure 3E, green puncta). Of note, phosphorylated β-catenin was largely cytoplasmic, while non-phosphorylated β-catenin (total) was primarily localized in the cell–cell junctions. Together, these results demonstrate that stimulation of human ECs with NS1 triggers the phosphorylation of β-catenin in a tissue-specific manner, consistent with our previous observations [19].

### 2.3. GSK-3β Is Required for NS1-Mediated Endothelial Hyperpermeability and Vascular Leak

Given the central role of phosphorylation of β-catenin in modulating the barrier function of epithelial cells [33], we assessed the contribution of GSK-3β, which phosphorylates S33/S37/T41, to flavivirus NS1-induced endothelial hyperpermeability in vitro and vascular leak in vivo. We utilized a small peptide competitive inhibitor of GSK-3β (GSK-3β Peptide Inhibitor, GPI) [41]. Using TEER as a measure of the integrity of the endothelial barrier, we demonstrated that the peptide inhibitor alone did not interfere with endothelial permeability, but that GPI was sufficient to inhibit DENV and ZIKV NS1-mediated hyperpermeability in a dose-dependent manner in HBMEC (Figure 4A,B) and HUVEC (Figure 4C,D) monolayers. Further, in a murine model of dermal endothelial permeability, in which fluorescently labeled dextran enables quantitation of local vascular permeability in vivo [9,22], GPI was tested in the presence of 15 µg of DENV NS1. GPI significantly reduced the DENV NS1-induced vascular leak in comparison to the NS1 + vehicle treatment without inducing vascular leak alone, in comparison to the PBS control (Figure 4E,F). These results support the critical contribution of GSK-3β to DENV NS1-induced permeability both in vitro and in vivo in relation to the phosphorylation of β-catenin, which culminates in mislocalization of IJC proteins.

## 3. Discussion

In this study, we demonstrated that NS1 proteins from closely related flaviviruses trigger endothelial hyperpermeability in an EC-type-dependent manner through alterations in the normal distribution of VE-cadherin and β-catenin, two important proteins of the IJC that are critical for maintenance of EC barrier function. This phenomenon occurred via clathrin-mediated endocytosis of VE-cadherin and phosphorylation of β-catenin, leading to their movement out of the IJC, which resulted in increased endothelial permeability and vascular dysfunction. Further, we demonstrated a critical role for GSK-3β, a cellular kinase that contributes to the phosphorylation of β-catenin, leading to the destabilization of the IJC complex during NS1-mediated endothelial hyperpermeability and vascular leak. A proposed model of this potential mechanism is depicted in Figure 5.

Endothelial cell-to-cell junctions are dynamically regulated structures comprised of TJ/AJ proteins whose expression and localization are critical to cell barrier function, regulating the paracellular transport of polar solutes and macromolecules into the tissues from the blood stream under distinct physiological and pathophysiological conditions [42,43]. The IJC is actively remodeled in response to multiple extracellular stimuli to transiently increase or decrease endothelial permeability [44]. Therefore, breakdown of the IJC results in permeability changes and barrier dysfunction, particularly under pathological conditions [45,46,47,48]. Displacement of IJC proteins such as ZO-1 and VE-cadherin can occur through multiple pathways [49,50]. In epithelial and endothelial cells, clathrin- and caveolae-mediated internalization are critical cellular mechanisms for the transport of IJC proteins from the plasma membrane to the cytoplasm, causing dissociation of the IJC [51]. Additionally, phosphorylation of some of the IJC proteins (e.g., β-catenin) has been shown to reduce the affinity of TJ/AJ protein–protein interactions, which results in destabilization of the TJ/AJ complexes and barrier dysfunction [33,52].

During viral infections, the modulation of IJCs has been described as a mechanism for viral dissemination and increased viral pathogenesis [53]. In many flavivirus infections, especially DENV infection, a hallmark of viral pathogenesis leading to severe disease is alterations in the barrier function of the endothelium, resulting in transient vascular leak and hypotension that can lead to shock and potentially death [2,54,55,56]. DENV, WNV, JEV, and ZIKV can alter the integrity of TJ/AJ proteins at the blood–brain and placental barriers [23,57,58,59,60,61,62,63,64,65,66]. The altered expression and/or distribution of IJC proteins such as VE-cadherin, ZO-1, and β-catenin has been described in in vitro assays with epithelial and endothelial cells after DENV or other flaviviral infections [59,60,61,62,63,64,65,67,68,69,70,71], stimulation of human ECs with soluble factors circulating in sera from dengue patients [72,73,74] or secreted from DENV-infected cells [18,75], as well as in DENV and ZIKV NS1-stimulated umbilical vein ECs, ex vivo human chorionic explants, and dermal ECs [10,23,39,40,58,76,77]. Importantly, IJC proteins have been found to circulate in the serum of dengue patients undergoing severe plasma leakage, and alterations of the IJC have been proposed as potential biomarkers for disease prognosis [15,22,78].

In this study, we identified that NS1 treatment of human ECs changed the distribution of VE-cadherin and β-catenin at cell-to-cell contacts without significantly altering the expression level of these pivotal IJC proteins. This phenomenon coincides with the peak of EC hyperpermeability (decreased TEER) occurring in EC monolayers exposed to soluble NS1 proteins between 6 and 8 h post-treatment, suggesting that altered distribution but not altered expression of IJC proteins may contribute to the transient peak of EC barrier dysfunction occurring during flavivirus infections. One limitation of our study is that IJC protein distribution and expression were only analyzed at a single time point, 6 hpt, when increased endothelial permeability peaks in vitro. Thus, although no changes in IJC protein expression were detected at this time point, it is possible that alterations may be observed at other time points. Further studies analyzing the kinetics of IJC protein distribution and expression after human EC exposure to flavivirus NS1 proteins will provide a more complete picture of the dynamics of NS1-medated barrier dysfunction.

Flavivirus NS1 proteins have been shown to selectively bind to and alter the permeability of human ECs and cause tissue-specific vascular leakage in vivo, reflecting the pathophysiology of each flavivirus [9,19,22,23]. For instance, NS1 from neurotropic flaviviruses, such as WNV, JEV, DENV, and ZIKV, can cause endothelial dysfunction in human brain ECs, and those flaviviruses in which transplacental infection has been described, such as DENV and ZIKV [79,80,81,82,83], the corresponding NS1 caused barrier dysfunction in umbilical vein EC used as a proxy for placental tissues. Mechanistically, this phenomenon is driven at the stage of binding and internalization of NS1 into ECs, which in turn triggers flavivirus-conserved steps, such as the activation of EC-intrinsic pathways (e.g., heparanase, sialidase), that result in the disruption of the EGL on the surface of ECs, contributing to vascular leak [8,19]. A complete picture of which amino acids dictate tissue-specific NS1 function is still lacking, but previous structural studies have identified three distinct domains of flavivirus NS1, namely, β-roll, wing, and β-ladder [84]. Our current and previous work highlight both conserved residues within the wing and β-ladder domains that all NS1 proteins require to trigger barrier dysfunction, as well as divergent residues within the wing domain that are required for tissue specificity [85]. Further, we generated chimeras and site-specific mutants of DENV, ZIKV, and WNV NS1 proteins that demonstrated the capacity to modulate EC tropism by swapping the wing and β-ladder domains. Specifically, we identified a three-amino acid motif of DENV that confers EC-binding tropism [86]. This is an important area of active investigation, which will improve understanding of the NS1 determinants that trigger EC barrier dysfunction.

While specific host factors required for NS1-medited IJC disruption are still obscure, previous studies have shown that secreted NS1 proteins from DENV and ZIKV disrupt the IJC (e.g., β-catenin and ZO-1) through induction of macrophage migration inhibitory factor (MIF) and matrix metalloproteinase-9 (MMP-9) [10,23,39], or through activation of human ECs via the inflammatory stress-sensing p38 MAPK pathway and/or Angiotensin 1/2 receptor signaling, which were required for NS1-triggered reduction of barrier integrity of human EC monolayers in vitro [40,76]. Here, we observed that treatment of EC monolayers with DENV and ZIKV NS1 proteins resulted in colocalization of VE-cadherin with clathrin heavy chain, suggesting its internalization via clathrin-mediated endocytosis. The phenomenon of EC barrier dysfunction was accompanied by increased staining of clathrin heavy chain proteins in HUVEC monolayers exposed to DENV and ZIKV NS1 proteins but not in the untreated ECs or ECs treated with WNV NS1. NS1 internalization in infected and non-infected cells, particularly via clathrin-mediated endocytosis, has been described as a cellular mechanism required for NS1-induced endothelial hyperpermeability, EGL degradation, and in DENV infection [22,40,76,87].

Additionally, we describe the phosphorylation and disassembly of β-catenin from the cadherin–catenin complex, correlating with the requirement of GSK-3β for NS1-mediated endothelial dysfunction, and its mislocalization in NS1-treated cells. GSK3 is a serine/threonine protein kinase that exists in two isoforms encoded by two different genes, GSK3α and GSK3β, with numerous well-documented roles in distinct cellular processes, including innate immune modulation and glucose metabolism [35,88]. In the past two decades, the recognized contribution of GSK3 kinases in human diseases including diabetes, cancer, and inflammation, as well as many viral infections and pathogenesis, has increased dramatically [36,89]. Of note, GSK3 has been identified as a host factor required for replication of SARS-CoV-1 and -2 [90,91], influenza [92], hepatitis C [93], dengue [94], and other viruses [95,96]. Yet, the role of GSK3 in mediating flavivirus pathogenesis related to the modulation of endothelial barrier function by secreted NS1 proteins has not been addressed. Here, given the pivotal role of GSK3 in regulating the fate of β-catenin in response to distinct external stimuli, we showed in vitro and in vivo that a specific inhibitor of GSK-3β activity significantly reduced NS1-mediated endothelial hyperpermeability and prevented NS1-mediated vascular leak, supporting the contribution of GSK3 kinases to NS1-induced pathogenesis. One caveat to our data is the potential for off-target effects of inhibitors such as the GSK3 peptide inhibitor to alter NS1-mediated IJC disruption. Thus, future genetic investigations into the role of GSK-3B are needed to further probe the role of this kinase in NS1-medated EC dysfunction.

Overall, this work provides new insights into the endothelial, cell-intrinsic mechanisms that contribute to endothelial hyperpermeability triggered by NS1 proteins from clinically important human flaviviruses such as DENV, ZIKV, WNV, and JEV. It confirms the role of the IJC as highly organized cellular structures that maintain the homeostasis of the endothelium and supports previous findings implicating the IJC in the pathogenic processes of DENV and other flavivirus NS1 proteins. Along with previous data, our results together suggest that the mislocalization of AJ proteins from cell-to-cell contacts in human ECs induced by NS1 is a complex multifactorial phenomenon involving several cellular mechanisms that act together to cause endothelial hyperpermeability and vascular leak. The kinetics and dynamics of this phenomenon are yet to be defined and future studies will define the spatial–temporal relationships between host factors and NS1.

## 4. Materials and Methods

### 4.1. Ethics Statement

All in vivo experiments were performed following the guidelines of the American Veterinary Medical Association and the Guide for the Care and Use of Laboratory Animals from the National Institutes of Health and were pre-approved by the University of California (UC) Berkeley Animal Care and Use Committee (protocol AUP-2014-08-6638) as previously described [9,11,19].

### 4.2. Mice

Six-to-eight-week-old wild-type C57BL/6 (B6) mice were obtained from the Jackson Laboratory. All mice were bred and maintained in specific pathogen-free conditions at the animal facility at UC Berkeley. A mix of male and female six-to-eight-week-old mice were used in all experiments. Mice were housed in a controlled-temperature environment on a 12-h light/dark cycle, with food and water provided *ad libitum*.

### 4.3. Cell Culture

All human endothelial cells used in this study were maintained following standard protocols as previously described [9,19,22]. Human umbilical vein ECs (HUVEC) were a kind gift from Melissa Lodoen at the University of California, Irvine. HUVECs are primary endothelial cells obtained from a single female donor (Lonza). HUVECs were propagated (passages 5–10) and maintained in endothelial cell basal medium-2 (500 mL) supplemented with FBS (5%), hydrocortisone (0.2 mL), R3-IGF-1 (0.5 mL), ascorbic acid (0.5 mL), hEGF (2 mL), and gentamicin-1000 (0.5 mL), as per the manufacturer’s specifications (Lonza). Human brain microvascular endothelial cells (HBMECs) were donated by Ana Rodriguez at New York University and maintained at low passages (passages 8–10) using endothelial cell medium (500 mL) supplemented with FBS (5%), 5 mL of endothelial cell growth supplement (ECGS, Cat. No. 1052), and 5 mL of penicillin/streptomycin (ScienCell Research Labs). All human ECs were tested to be mycoplasma-free, kept at low passage at 37 °C in humidified air with 5% CO_2_, and routinely passaged prior to reaching 100% confluence.

### 4.4. Recombinant NS1 Proteins

Recombinant NS1 proteins from DENV-2 (Thailand/16681/84), ZIKV (Suriname Z1106033), WNV (NY99), JEV (SA-14), and YFV (17D) were commercially acquired from the Native Antigen Company (Oxfordshire, UK). These proteins were produced in HEK293 cells with a purity greater than 95%, composed mostly of oligomeric forms as confirmed by silver staining and western blot assays [19], and certified by the manufacturer to be free of endotoxin contaminants.

### 4.5. Monoclonal Antibodies, and Inhibitors

For staining for AJ proteins, the following monoclonal antibodies (mAbs) were used: anti-β-catenin (CAT-5H10, mouse IgG, Thermo Fisher Scientific, Waltham, MA, USA) and anti-phospho β-catenin (S33/S37/T41) (Rabbit, Cell Signaling Technology, Danvers, MA, USA). Secondary antibodies used were goat anti-mouse IgG conjugated to Alexa Fluor 647 (Abcam, Cambridge, UK) and goat-anti-rabbit IgG conjugated to Alexa 568 (Abcam). The same panel of mAbs was used for Western blotting along with the anti-GAPDH mAb (Rabbit, Santa Cruz Biotechnology, Santa Cruz, CA, USA). For chemical inhibition of the GSK-3β activity, we used a cell-permeable GSK-3 peptide inhibitor (Millipore Sigma™ Calbiochem™, Steinheim, Germany) at concentrations (0.2–40 μM) that do not affect the cell viability, as previously demonstrated [41].

### 4.6. Trans-Endothelial Electrical Resistance (TEER)

The effect of recombinant flavivirus NS1 proteins on endothelial permeability was evaluated by measuring TEER of EC monolayers grown on a 24-well Transwell polycarbonate membrane system (Transwell^®^ permeable support, 0.4 μm, 6.5 mm insert; Corning Inc., Corning, NY, USA), as previously described [8,19]. Briefly, HUVECs or HBMECs (80–90% confluency) cultured in vented 75 cm^2^ flasks (Corning^®^) were detached using a combination of three washing steps using sterile 1× PBS supplemented with EDTA (2 mM) (lifting buffer), followed by two additional washing steps (~30 s total) using a solution of trypsin-EDTA (0.25%) (GIBCO, Thermo Scientific, Waltham, MA, USA). Detached cells were resuspended using fresh culture medium and then counted using a tissue-culture hemocytometer. A total of 1 × 10^5^ cells in 300 uL of medium were seeded on the apical side of Transwell inserts (top chamber) placed in a 24-well plate containing 1.5 mL of EC culture medium (basolateral side). Transwells containing ECs were incubated at 37 °C and 5% CO_2_ for 3 days, and 50% of culture medium was changed in each well every day post-seeding. Cells were grown until TEER values between 150 and 180 Ohms (Ω) were reached, depending on cell type, indicating 100% cell confluency. Individual flavivirus NS1 proteins (5 μg/mL, 1.5 μg total protein) were then added to the apical side of the Transwell insert (top chamber, 300 μL) containing the cell monolayer. TEER values, expressed in Ohms (Ω), were collected at sequential 2-h time-points over 3–11 h following the addition of test proteins using an Epithelial Volt Ohm Meter (EVOM) with a “chopstick” electrode (World Precision Instruments). Inserts with no cells containing medium alone were used as negative controls to calculate the baseline electrical resistance. Endothelial permeability was expressed as relative TEER, which represents a ratio of resistance values (Ω) as follows: (Ω experimental condition—Ω medium alone)/(Ω non-treated endothelial cells—Ω medium alone).

### 4.7. Fluorescence Microscopy

For imaging experiments, HUVECs and HBMECs were grown on coverslips and imaged on a Zeiss LSM 710 Axio Observer inverted fluorescence microscope equipped with a 34-channel spectral detector. Images acquired using the Zen 2010 software (Zeiss, Jena, Germany) (708.49 × 708.49 μM) were processed and analyzed with ImageJ software [97]. For representative pictures (RGB format), an area of 163.12 × 178.77 μm containing ~10–30 cells was used. To assess the effect of flavivirus NS1 on integrity of the endothelial architecture, the distribution of VE-cadherin and β-catenin was examined on confluent EC monolayers treated with distinct flavivirus NS1 proteins, as indicated in each figure legend. EC monolayers were fixed with 2% paraformaldehyde (PFA) and cold methanol (1 mL) at 6 hpt. A permeabilization step was performed using saponin (0.2%) in blocking buffer (3% BSA, 1% FBS in PBS 1×) for 30 min at room temperature. Primary antibodies were added and incubated overnight at 4 °C in PBS (1×), and detection was performed using secondary species-specific anti-IgG antibodies conjugated to Alexa fluorophores (568 and 647). Nuclei were stained using Hoechst. Mean fluorescence intensity (MFI) values for β-catenin staining on human ECs treated with NS1 or controls were obtained from individual RGB–grayscale-transformed images (n = 3). All images were processed, edited, and analyzed using ImageJ software, as previously described [8,9,19,22,23,97].

### 4.8. Western Blot Analyses

For protein expression, confluent EC monolayers (~1 × 10^6^ cells/well, 6-well tissue culture-treated plates) were treated with the five flavivirus NS1 proteins (DENV, ZIKV, WNV, JEV, and YFV) separately (5 μg/mL), and at 6 hpt, cell monolayers were scraped on ice using RIPA lysis buffer (50 mM Tris (pH 7.4), 150 mM NaCl, 1% (*v/v*) Nonidet-P40, 2 mM EDTA, 0.1% (*w/v*) SDS, 0.5% Na-deoxycholate, and 50 mM NaF) supplemented with a complete protease inhibitor cocktail (Roche, Basel, Switzerland). After total protein quantification using a bicinchoninic acid (BCA)-based colorimetric assay (Pierce BCA Protein Assay Kit, Thermo Scientific, Waltham, MA, USA), 10 μg of total protein per sample was boiled and placed in reducing Laemmeli buffer and separated by 4–20% gradient SDS-PAGE. After immunoblotting using specific primary antibodies for β-catenin and GAPDH (used as housekeeping protein control) and secondary species-specific anti-IgG antibody conjugated to Alexa 680 or Alexa 750, protein detection and quantification was carried out using the Odyssey CLx Infrared Imaging System (LI-COR, Biosciences, Lincoln, NE, USA). Relative densitometry represents a ratio of the values obtained from each experimental protein band over the values obtained from the loading controls (GAPDH) after subtracting the background from both using Image Studio Lite V 5.2 (LI-COR Biosciences, Biosciences, Lincoln, NE, USA) [8,19].

### 4.9. ELISA

Instant one step ELISA for human β-catenin (PhosphoTracer β-Catenin Total ELISA Kit) on adherent cells was performed following the manufacturer’s instructions (Thermo Fisher Scientific, USA). Briefly, confluent HUVEC monolayers cultured in 48-well plates were treated with distinct flavivirus NS1 proteins (5 μg/mL), and cell protein lysates were collected 6 hpt following the manufacturer’s protocol. The amount of total and phosphorylated (S45) β-catenin per well was determined by fluorogenic quantification using a Spectra Max (M3) microplate reader (Molecular Devices, San Jose, CA, USA) equipped with a dual monochromator spectrofluorometer system that provides excitation and emission wavelength selection between 250–850 nm, as previously described [98]. The amount of phosphorylated (S45) β-catenin was expressed as the percentage of the total β-catenin as 100%.

### 4.10. Localized Vascular Leak Murine Model Assay

In vivo NS1-induced endothelial hyperpermeability was measured using a rodent model of localized vascular leak, as previously described [9]. Briefly, the dorsal hair area of 6-week-old female WT C57BL/6 mice (Jackson Labs) was depilated 3–4 days prior to each experiment. On the day of the experiment, mice were anesthetized with isoflurane and injected into the shaved dorsal dermis with 50 μL of four different treatments as follows: 1× PBS plus vehicle (1 μL of DMSO in 50 μL PBS1×), GSK3β inhibitor alone (10 μg in 1 μL DMSO plus 50 μL 1× PBS), DENV2 NS1 (15 μg in 1 μL plus 50 μL 1× PBS) alone, and the mixture of GSK3β inhibitor plus DENV2 NS1 (15 μg NS1 plus 10 μg GSK3β inhibitor in 50 μL 1× PBS). Local vascular leakage at the injection spot was quantified and expressed as the mean fluorescence intensity of 10-kDa dextran conjugated with Alexa Fluor 680 (1 mg/mL; Sigma) delivered via retro-orbital (RO) injection. Two hours post-injection, mice were euthanized using isoflurane, and the dorsal dermis was removed and placed in Petri dishes. Tissues were scanned using a fluorescent detection system (LI-COR Odyssey CLx Imaging System) at a wavelength of 700 nm, and leakage in a 13-mm diameter circle surrounding the sites of injection was quantified using Image Studio software (LI-COR Biosciences, Lincoln, NE, USA).

### 4.11. Data Analyses and Statistics

All experimental conditions were repeated three times, and images are representative of these experiments. Statistical differences between groups were determined by Ordinary one-way ANOVA and considered significant with *p*-values <0.05. Comparison between MFI, ELISA, and densitometry data was conducted using multiple *t*-tests. All data analyses, statistics, and graphs were performed and generated using GraphPad Prism v6.07.

## Figures and Tables

**Figure 1 pathogens-11-00615-f001:**
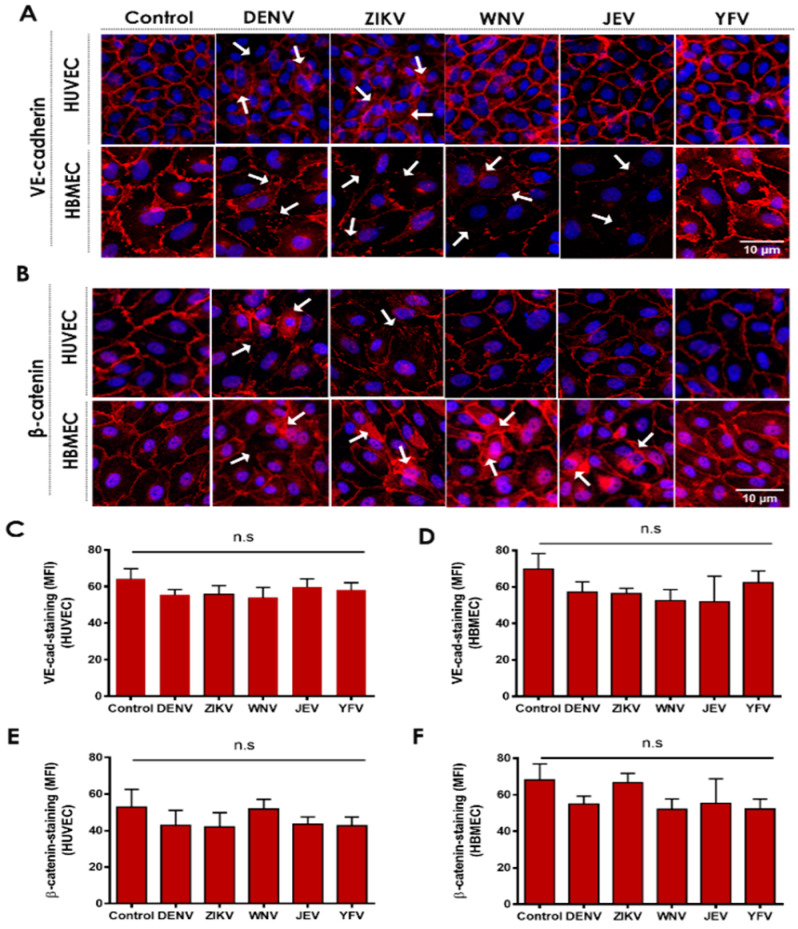
Flavivirus NS1 proteins induce mislocalization of IJC proteins in human endothelial cells. Confluent monolayers of HUVECs (*top panel*) and HBMEC (*lower panel*) were treated individually with five different NS1 proteins from DENV, ZIKV, WNV, JEV, and YFV (5 μg/mL). Monolayers were fixed and processed 6 h post-treatment (hpt) by IFA staining of (**A**) VE-cadherin and (**B**) β-catenin (both in *red*). Untreated (no NS1) cells were used as a control for normal distribution/localization of VE-cadherin and β-catenin. Nuclei are depicted in blue (*Hoechst*). Mean fluorescence intensity (MFI) analysis of (**C**,**D**) VE-cadherin and (**E**,**F**) β-catenin staining (IFA) per NS1 treatment and control treatment of (**C**,**E**) HUVEC and (**D**,**F**) HBMEC are shown. Images are representative of three independent experiments (area: 29.16 mm^2^). White arrowheads indicate places of VE-cadherin and β-catenin mislocalization, suggesting disruption of the IJC. Scale bar: 10 μm. Magnification: 20×.

**Figure 2 pathogens-11-00615-f002:**
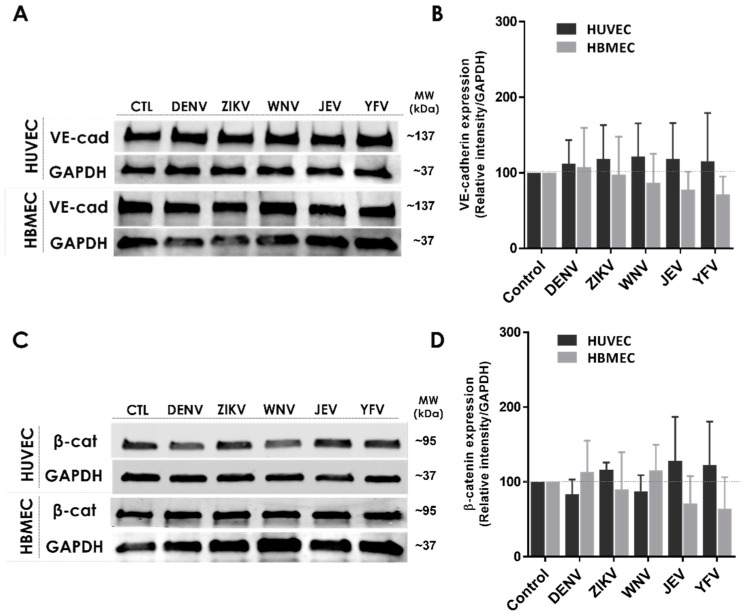
Flavivirus NS1 proteins do not change the expression of IJC proteins in human endothelial cells. (**A**,**C**) Western blot analyses of the expression of VE-cadherin and β-catenin 6 hpt with five distinct flavivirus NS1 proteins as indicated. (**B**,**D**) VE-cadherin and β-catenin protein expression examined by densitometry analyses of the Western blot bands of human ECs treated with NS1 proteins and control cells. VE-cadherin (~137 kDa) and β-catenin (~95 kDa) at 6 hpt. GAPDH (~37 kDa) was used as a protein loading control. The fluorescence intensity of the protein bands, corresponding to the molecular size (kDa) of VE-cadherin and β-catenin in each human EC, was determined by ImageJ Studio analyses and plotted as the ratio of their relative intensity vs. GAPDH from the same experimental condition. Images are representative of three independent experiments.

**Figure 3 pathogens-11-00615-f003:**
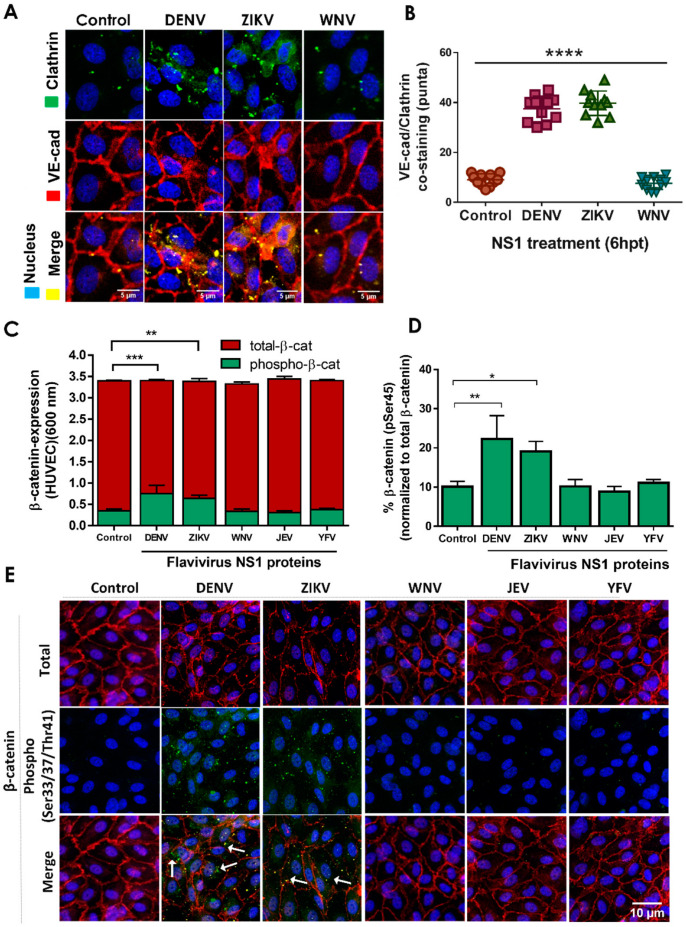
Treatment of HUVEC monolayers with DENV and ZIKV NS1 proteins but not WNV NS1 increases the colocalization of VE-cadherin with clathrin and the phosphorylation and cytosolic accumulation of β-catenin. Confluent monolayers of HUVEC were individually treated with five different NS1 proteins from DENV, ZIKV, WNV, JEV and YFV (5 μg/mL), and 6 hpt, cell monolayers were processed by (**A**,**B**) IFA for co-staining of VE-cadherin and clathrin heavy chain, (**C**,**D**) commercial ELISA for detection of total/phosphorylated β-catenin (S45), or (**E**) IFA for co-staining of total β-catenin (red) and phosphorylated β-catenin (S33, S37, T41) (green). Images are representative of three independent experiments. White arrows indicate VE-cadherin co-localization with clathrin heavy chain proteins (**A**) and the cytosolic accumulation of phosphorylated β-catenin (green) (**E**). Merge is indicated in yellow. IFA co-staining analyses were performed by using RBG images and ImageJ software analysis. For both assays, untreated (no NS1) cells were used as a control for normal β-catenin expression/distribution and phosphorylation in HUVEC monolayers. Images are representative of three independent experiments (area: 29.16 mm^2^). White arrowheads indicate colocalizing puncta of VE-cadherin and clathrin proteins. Scale bar: 10 μm. Magnification: 20×. Significant differences as follows: *p* < 0.05 *, *p* < 0.01 **, *p* < 0.001 ***, *p* < 0.0001 ****.

**Figure 4 pathogens-11-00615-f004:**
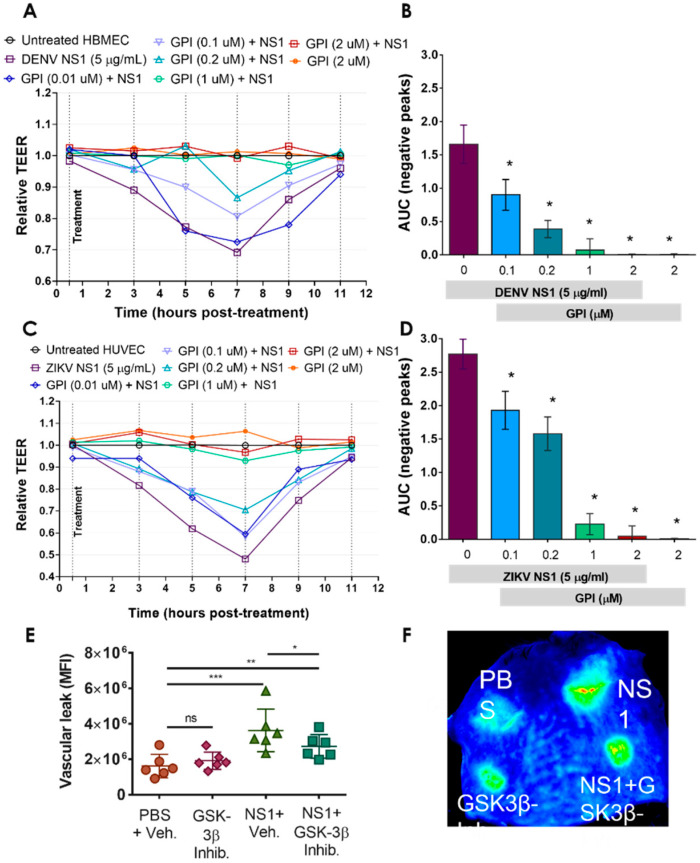
A GSK-3β inhibitor prevents DENV and ZIKV NS1-induced increased permeability in human endothelial cells in vitro and reduces vascular leak in vivo. (**A**–**D**) Confluent monolayers of HBMEC (**A**,**B**) and HUVEC (**C**,**D**) grown on Transwell semipermeable membranes were treated individually with DENV or ZIKV NS1 proteins in the presence of different concentrations (μM) of a GSK-3β peptide inhibitor (GPI). TEER values were collected for 48 hpt to evaluate the barrier function of HBMEC and HUVEC. Untreated (no NS1) cells were used as a control of normal barrier function of human ECs. HBMEC and HUVEC monolayers treated only with the GSK-3β inhibitor were used as a control to assess nonspecific cell cytotoxicity resulting from the inhibitor alone. Cell treatment with NS1 proteins alone (plus DMSO vehicle with no inhibitor) were used as a positive control for increased endothelial permeability. AUC, negative area under the curve. (**E**,**F**) Intradermal injection of the DENV NS1 protein in the presence or absence of the GSK-3 inhibitor GPI (40 μM) was performed on the dorsal skin of WT C57BL/6 mice (n = 6). Fluorescently labeled dextran (10 kDa) was used to detect fluid extravasation at the site of injection as an indicator of vascular leak. The dermis from each mouse was collected and processed 2 h post-injection. Mean intensity values were collected using a LI-COR Odyssey CLx Imaging System at a wavelength of 700 nm. The image in E is representative of a mouse dorsal skin scanned using the LI-COR Odyssey CLx Imaging System and was performed three times independently. PBS (*top left*); DENV2 NS1 (15 μg) (*top right*); GPI (40 μM) (*bottom left*); and DENV2 NS1 plus GPI (*bottom right*). Significant differences as follows: *p* < 0.05 *, *p* < 0.01 **, *p* < 0.001 ***.

**Figure 5 pathogens-11-00615-f005:**
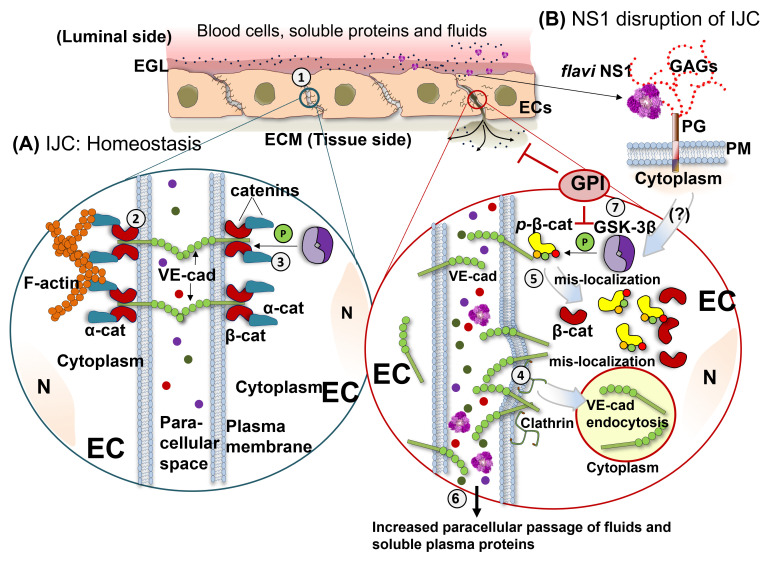
Mechanistic overview of flavivirus NS1-induced intercellular junction complex dysfunction. (**A**) Intercellular junction complexes (IJC) are composed of the tight and adherens junction (TJ/AJ) protein complexes and are located at cell-to-cell contacts (1). IJCs are major components required for maintaining endothelial barrier integrity, which regulates the passage of fluids and soluble molecules between the bloodstream and underlying tissues. Under homeostatic conditions, the AJ complex forms tight barriers between cells containing VE-cadherin, which is anchored to the cell cytoplasm via association of its cytoplasmic domain with β-catenin, α-catenin, and cytoskeletal components (2). Localization of β-catenin to AJs, and thus IJC barrier integrity, is regulated via several phosphorylation events mediated by kinases, including GSK-3β (3). Our current study shows that tissue-specific flavivirus NS1-mediated endothelial dysfunction is associated with mislocalization of the IJC proteins, VE-cadherin, and β-catenin (**B**). Our study reveals mechanistic details of AJ disruption, demonstrating that NS1 treatment of endothelial cells is associated with clathrin-mediated endocytosis of VE-cadherin (4) and phosphorylation of β-catenin (5), which is known to destabilize the β-catenin/VE-cadherin complex, resulting in endothelial barrier dysfunction (6). In agreement with these observations, our study demonstrates that a GSK-3β-specific peptide inhibitor (GPI) is sufficient to abrogate NS1-mediated endothelial permeability and vascular leak (7). Thus, our study provides new insights by which TJ/AJ barrier integrity is disrupted by NS1. Abbreviations: EGL (endothelial glycocalyx layer); ECs (endothelial cells); ECM (extracellular matrix); IJC (intercellular junction complex); N (nucleus); β-cat (β-catenin); α-cat (α-catenin); GAGs (glycosaminoglycans); PG (proteoglycan); PM (plasma membrane); VE-cad (VE-cadherin). *p*-β-cat (phosphorylated β-catenin).

## Data Availability

All data generated or analyzed during this study are included within this published article.

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
