# Peer review of "Flavivirus NS1 Triggers Tissue-Specific Disassembly of Intercellular Junctions Leading to Barrier Dysfunction and Vascular Leak in a GSK-3β-Dependent Manner"

_pathogens, 2022, doi:10.3390/pathogens11060615_

Round 1

Reviewer 1 Report

Puerta-Guerdo and collegues examined the effect on distribution, expression of NS-1 proteins of five Flaviviruses on two cell surface molecules expressen ont he surface of endothelial cells of the umbilical vein and brain. They showed that NS1 stimulation of human ECS from umbilical vein and brain induces redistribution of VE-cadherin and β-catenin molecules from cell-cell junctions, without changing their protein expression, by inducing their mislocalization from the IJC via clathrin-mediated endocytosis or phosphorylation by GSK-3β.

The results are interesting and novel help to understand better an important part of Flaviviral pathogenesis, way of viral entry to teh CNS. The English is good, the applied methods are appropriate, the figures help in understanding.

I suggest its acceptation after including the answers to the questions below into the paper.

I missed throughout the paper the Tick-borne encephalitis virus. Even if it is not present in the American continent one of the main arthropode-borne zoonotic threat to humans in Europe and North-Asia, causing thousands of hospitalized patients and hundreds of deaths annually. As an important member of the Flavivirus group, should be mentioned in line 38.

lines 131-135. Fig 1., The staining pattern of mislocalization of the two molecules was virus-and cell-type specific. Is it known which part/sequence of the flaviviral NS-1 proteins  is responsible for these differences?

 lines 145-147. – Stainings proved the mislocation but not degradation of the two IJC proteins. How long this mislocation lasts? Is not it possible that in a day these molecules regain their original position in the cell surfaces? Can not be this mislocation of molecules transient only?

lines 178-182- Is this means that increased colocalization of the two molecules were detectable only in case of DENV and ZIKAV? The other tested Flaviviruses were included in these tests, or these viruse did not altered the colocalization of VE-cadherin and clathrin on HUVEC ?

lines 187-192 - Why Zika and dengue seem to be more sufficent in altering the normal activity of IJC molecules? Why not the others?

line 233. The word ’Discussion’ is at very wrong place here. Must be transfer to lines 252-253, after the Figure 4. legend.

Do the authors think that these molecular changes in the IJ cells could be reason for flaviviral entry from the blood circulation to the CNS?

I have not checked the accuracy of references.

Author Response

Dear Reviewer.

We appreciate all your comments and suggestions.

Reviewer 2 Report

Puerta-Guardo and colleagues investigated the effect of NS1 proteins from different flaviviruses on the integrity of endothelial barrier in the models of human umbilical cord and brain endothelial cells. They showed that different NS1 induce redistribution of intercellular junction proteins in a model-specific pattern. NS1 of Dengue and Zika viruses trigger relocalization of VE-cadherin  from the plasma membrane via clathrin-dependent endocytosis and phosphorylation of b-catenin. Treatment of cells with an inhibitor of GSK-3B prevented NS-1-induced disruption of endothelial cell monolayer and the in vivo vascular leakage in a murine model, suggesting that this kinase could be a target to prevent NS1-induced hemorrhagic effect. 

The results are convincing and well presented, I have only minor suggestions for the improvement of the manuscript:

Fig. 3A. Please provide higher magnification (single cell) insets for individual channels. In the control cells the number of clathrin-positive dots is apparently significantly lower, can you comment on how NS1 could increase clathrin-dependent endocytosis?

Fig 3A and E show suggestively similar patterns of the distribution of VE-cadherin and phosphorylated b-catenin. An addition of a clathrin staining to phosphocatenin may extend the conclusion about the mechanism of cell junction protein redistribution upon NS1 treatment.

Fig. 4. The results presented do not unequivocally establish the role of GSK-3B in NS1-induced permeability and can be confounded by a possible interaction of GPI peptide with NS1 preventing the engagement of cellular targets by NS1s. This should be added to the Discussion.

Author Response

(The authors gave the same response as above.)
